# The Anterolateral Ligament of the Knee in Pediatric Patients: What Do We Know? A Scoping Review

**DOI:** 10.3390/jfmk8030126

**Published:** 2023-09-04

**Authors:** Ludovico Lucenti, Gianluca Testa, Marco Montemagno, Marco Sapienza, Arcangelo Russo, Fabrizio Di Maria, Claudia de Cristo, Vito Pavone

**Affiliations:** 1Department of General Surgery and Medical Surgical Specialties, Section of Orthopaedics and Traumatology, Policlinico Rodolico-San Marco, University of Catania, 95123 Catania, Italy; ludovico.lucenti@gmail.com (L.L.); gianpavel@hotmail.it (G.T.); docmontemagno@gmail.com (M.M.); marcosapienza09@yahoo.it (M.S.); fdimaria95@gmail.com (F.D.M.); decristo.claudia@gmail.com (C.d.C.); 2Orthopaedic and Traumatology Unit, Umberto I Hospital, 94100 Enna, Italy; arcangelo.russo@unikore.it

**Keywords:** knee, anterolateral ligament, ALL, ligament injury, pediatrics, anterolateral capsule

## Abstract

The knowledge on the anatomy, function and biomechanics and the role of surgical procedures on the anterolateral ligament (ALL) of the knee is still controversial. Only a few papers have examined the ALL in children. The aim of this review is to analyze all the available literature about ALL in the pediatric population. Following the PRISMA criteria, the literature was systematically reviewed, examining all the articles about ALL in pediatric patients. Eight articles were involved in this study. Five cadaveric studies, two diagnostic studies, and one cross-sectional study were found. The identification of the ALL is not always possible in diagnostic studies using magnetic resonance (MRI) or in dissecting specimens. A high variability in the presence of the ligament and in its origin and insertion were found among the studies. It is more difficult to identify the ligament in younger patients than in older children, suggesting that its presence may develop at some point during the growth. Further studies are needed for a detailed knowledge of the ALL.

## 1. Introduction

The anterolateral ligament (ALL) of the knee was originally described (but not named) in 1879 by the French surgeon Paul Segond [1], who stated the existence of a “fibrous band” in the lateral portion of the knee joint.

After its identification, the name “anterolateral ligament” was first used in 2012, once again in France, in cadaveric specimens [2]. The authors described the anatomy of the ligament, stating its origin on the lateral femoral condyle and the insertion on the lateral meniscus and tibial plateau, someplace posteriorly to Gerdy’s tubercle [2].

Subsequently, many authors tried to offer a full anatomical characterization of this apparent ligamentous structure. Some studies reported that the femoral origin of the ALL is variable [3,4,5], while the insertion was described (and afterward confirmed in different research papers) in the mid-lateral portion of the meniscus and the tibia, in the middle between Gerdy’s tubercle and the fibular head [6].

The function and biomechanics of the ALL are still controversial. Previously, Segond reported that this “pearly, resistant, fibrous band showed extreme amounts of tension during forced internal rotation”.

Since the 1970s, the idea that anterolateral rotatory instability was imputable not simply to ACL tears but also to the structures of the anterolateral complex, started to gain popularity.

Hughston et al., showing various patterns in rotational instability, illustrated a structure, the “mid-third lateral capsular ligament”, that was closely united with the lateral meniscus and was implicated in rotational instability [7].A few more studies mentioned this structure between 1970s and 1980s [8,9].

More recently, many studies confirmed that in knee flexion, the ALL resists internal rotation [10,11], while the role of the ALL in anterior stability is not fully understood [12,13].

The anterolateral knee maintains stability for anterior translation and anterolateral rotation of the knee. Some authors stated that is not a single structure that controls rotational stability but rather various structures work together: the ALL, the iliotibial band (ITB), the capsulo-osseous layer, and the Kaplan fibers (connections between the iliotibial band and the distal femur) [14].

Many recent studies have detected the ALL using diagnostic imaging (mainly MRI), cadaveric dissections, and surgically (using arthrotomy or arthroscopy) [15,16,17]. On the other hand, probably due to differences in anatomic dissection techniques, different authors denied the existence of this ligament, accentuating the relevance of other structures, like the deeper part of the iliotibial band (ITB) and the anterolateral capsule [18,19].

The identification of the ALL is sometimes very difficult, and detection of ALL tears is even more difficult.

Different authors thought that some ACL failures (re-rupture or clinical failure) may be due to ALL incompetence [20]. For this reason, the role of ALL reconstruction has become a subject of interest in many studies [21,22,23].

This ligament is not only significant in biomechanics but is also very important for its role in surgical procedures. In adults, combined ACL reconstruction with lateral extra-articular tenodesis (LET) or ALL reconstruction procedures are always more popular and show promising results [21,24,25]. Many reconstruction techniques have been proposed in the last decades, showing good outcomes in primary ACL injuries and ACL revisions.

The presence and the role of the ALL in the pediatric population is even more controversial compared to the adult population, and only a small number of authors have examined the ALL in children. In fact, because of the thinness and the poor identifiability of the ALL in children, its role in knee stability is uncertain. Furthermore, is not clear if the absence of the ALL may enhance the probability of an ACL tear in children who play sports [26].

The aim of this review is to analyze all the literature available about ALL in pediatric patients.

## 2. Materials and Methods

### 2.1. Search Selection

This scoping review was completed following the recommendations of the PRISMA Extension for Scoping Reviews (PRISMA-ScR) [27].

The aim was to find all the available literature about the ALL in children.

One medical electronic database (PubMed) was explored by a single author (LL) on the 25th of May 2023. The research keywords used were “((paediatric) OR (pediatric) OR (children) OR (kids)) AND ((knee) AND (anterolateral ligament)).” MesSH terms were included. No temporal limitations were established. A total of n = 53 articles were found.

Studies providing any level of evidence about the anterolateral ligament of the knee in pediatric patients were considered eligible for this study.

### 2.2. Inclusion and Exclusion Criteria

All the articles about other topics, with inadequate scientific methods, or without an available abstract were not considered. Only articles written in English were included in this study. No duplicates were found.

### 2.3. Data Extraction

The main parameters in each article were collected including the aim of the study, sample, presence of a control group, sex, side, mean age, results, and type of study.

### 2.4. Quality Assessment

Three authors (C.d.C., G.T. and F.D.M.) individually read the selected articles and evaluated their quality. All the authors discussed the quality of the articles selected. The senior author (V.P.) intervened in doubt cases.

## 3. Results

After the first preliminary screening conducted by three authors (C.d.C., G.T. and F.D.M.), n = 12 articles were considered adequate for a reading of the full text. A precise full-text reading of the selected papers was completed by the same authors to verify their eligibility and n = eight articles were selected, which met the inclusion criteria

The senior author (V.P.) intervened if a reviewer had any doubt about the inclusion of a study. The selected articles and their references and the ineligible studies were re-evaluated and then debated by all the authors to reduce the risk of bias.

A PRISMA [28] flowchart is shown in Figure 1.

The reference lists in the chosen articles were examined. The selected articles [29,30,31,32,33,34,35,36] are summarized in Table 1.

The articles about the ALL in children that were included in this study were sometimes heterogeneous, but they could easily be classified into three main categories (cross-sectional, cadaveric and diagnostic studies) as follows:-Survey—Cross-sectional studies

Madhan et al. [30] aimed to verify surgeons’ experiences about anterolateral ligament reconstruction (ALLR) as well as lateral extra-articular tenodesis (LET) in the pediatric patients.

-Cadaveric/descriptive anatomic studies

In 2022, Radhawa et al. [29] evaluated nine cadaveric specimens analyzing the pediatric anterolateral knee anatomy. In particular, they analyzed the LCL origin, the popliteus origin, and the insertion of the iliotibial band (ITB) as well as their relationships with physeal structures and how this structural information might impact operating procedures for lateral extra-articular procedures.

Iseki et al. [31] explored 15 pediatric knee anterolateral capsules (ALCs) from specimens aged 6.3 ± 3.3 years old and did not detect a clear ligament using histology, immunohistochemistry, or gene expression evaluation.

Shea et al. [34] studied the incidence of the ALL in 14 preadolescent anatomic specimens with a mean age of 8 years (range 7–11 years) and stated that the presence of this ligament in pediatric specimens is lower than in adult specimens.

Helito et al. [35] examined 20 fetal cadaveric specimens aged 28.64 ± 3.20 weeks and concluded that the ALL is present during fetal development.

Shea et al. [36] dissected eight skeletally immature knee specimens of different ages (between 3 months and 10 years). They recognized the ALL only in one of the eight specimens. The authors stated that, considering this ligament is present in most adult specimens (as reported in other studies [37]), the ALL probably matures later in life.

-Diagnostic studies

Helito et al. [33] observed 363 knee MRIs and found the presence of the ALL in 69.4% of all the knees of pediatric patients evaluated. The authors reported a lower visualization of the ligament in younger patients and a higher visualization in older patients (almost 18 years old). The ALL was not seen in female patients younger than 7 years or in male patients younger than 6 years.

The ligament was mostly seen in coronal sequences.

Liebenstein et al. [32] observed 61 MRIs of teenagers with a mean age of 15 years (±2.3). The authors reported the presence of the femoral part of the ALL in 72.1% of the cases examined, while the meniscal part and the tibial part of the ALL were visible in 0% and 78.7% of the cases, respectively. The deep connections between the iliotibial tract (ITT) and the distal femur were detected in 62.3% of the cases.

## 4. Discussion

### 4.1. History

The first citation of the ALL was in 1879, when Segond described an avulsion fracture of the anterolateral margin of the tibia and a “pearly band extending in an oblique fashion from the femur inserting into the avulsed tibial bone”. In the 1970s, Hughston et al. [7] illustrated a structure intimately connected to the lateral meniscus and apparently implicated in rotational instability. Between the 1970s and 1980s, two other authors (Johnson and Terry) [8,9] described the same ligament; however, only Vieira in 2007 [38] stated that in the anterolateral portion of the knee, the capsule–osseus layer of the ITB works as an “anterolateral ligament”, which are terms used for the very first time by Vincent et al. [2]. An accurate anatomical description of the ligament as a distinct structure was then proposed by Claes [6] in 2013.

### 4.2. Anatomy, Histology, and Biomechanics

The anatomy of the ALL varies largely between individuals, especially regarding the femoral insertion [39]. The proximal origin of the ALL is usually located on the lateral femoral epicondyle, anteriorly to the lateral collateral ligament (LCL) insertion and proximally and posteriorly to the insertion of the popliteus tendon. The ALL and LCL are considered together as the “lateral collateral ligament complex”. The insertion of the ALL is located posteriorly to Gerdy’s tubercle and anteriorly to the fibular head [40]. Some fibers in the ligament continue into the lateral intermuscular septum of the thigh, some are united to the LCL, and some are linked to the lateral meniscus. Some different ideas about the ligament anatomy are reported in the literature. Some authors [41] stated that two bundles of the ligament exist (one superficial and one deeper). Some others reported a close correlation (anatomical and functional) between the ALL and ITB [42].

A histological examination helped to prove the presence of the ALL as a separate structure, individuating not only a capsular thickening but rather a well-organized connective tissue [35] made by type I collagen (90%) and fibroblasts [43]. The structure of the ALL is like the ACL regarding cellularity, characteristics, fiber orientation, and nuclei form [35].

The function of the ALL in knee biomechanics is still controversial, probably due to differences in dissection techniques used in different studies. Some authors stated that the ALL stabilize the knee in extension and internal rotation, but others showed that it stabilizes the joint only during flexion [10].

### 4.3. Identification of the Ligament

Recognizing the ALL is not always simple, and its existence is not always accepted in the literature. This disagreement is essentially due to how the studies and dissections were performed.

The ligament can be identified using diagnostic studies (mostly MRI) or arthroscopically. The identification of ALL tears is even more difficult.

MRI studies performed on ACL injuries showed an agreement of 40–80% in identifying ALL tears. These divergences are due to MRI quality and the criteria for selecting patients [44]. The best way to identify the structure is using a 3 T MRI with a 0.4mm slice and fat-suppressed acquisition [45].

Zein reported that it is possible to identify the ligament arthroscopically using a 30° scope through the anterolateral portal. In his study, he gave precise instructions on how to observe the ligament [46].

### 4.4. Surgical Techniques

Many procedures for the ALL are reported in the literature. Sometimes, ALL reconstruction (ALLR) is used as a synonym for lateral extra-articular tenodesis (LET) because both surgical techniques have the purpose of restoring the anterolateral stability [25]. However, the latter is only functional and not an anatomical reconstruction, while the ALLR points to an anatomical reconstruction of the ALL.

LET was initially utilized as unique procedure in ACL injuries, but it showed a high risk of failure. The combination of ACLR and LET is now again of interest, and more than 10 different LET techniques are described in the literature without a superiority of one procedure over the others [47].

Indications for ALLR include, but are not limited to, high-grade pivot shift on examination (3+), young age (<20 years old), high-demand athletes, Segond fractures, revision ACLR, and chronic ACL injuries (>12 months) [20].

One of the most used techniques for the ALLR consists of a reconstruction of the ACL and ALL using a semitendinosus graft (used for ACLR) combined with a gracilis tendon graft (used for ALLR) [21].

Many studies showed that fixing the graft used in ALLR distally to Gerdy’s tubercle and proximally and posteriorly to the distal meta-epiphyseal junction of the femur leads to an adequate isometry, even if this placement is not anatomical [48]. Some authors stated that the greatest agreement between anatomy and isometry is achieved when the ligament is fixed posteriorly and proximally to the lateral epicondyle in the femur and between the anatomical ALL insertion and Gerdy’s tubercle in the tibia. The ligament should be fixed in knee extension without tensioning the ligament in external rotation to prevent stiffness [14,21,48].

Post-operative rehabilitation protocols are usually the same as those used for isolated ACL reconstruction.

Not many studies were conducted about clinical outcomes. However, a good recovery was reported, and it seems that ACL revision reconstruction associated with ALL reconstruction reduces rotational laxity and shows a high rate of return to sports [21].

### 4.5. Overview of the Pediatric Population

The presence of the ALL is largely studied in adult patients. However, a few studies have been conducted about the existence of the ALL in the pediatric population. The studies can mainly be distinguished into two types: diagnostic studies and cadaveric studies.

Most diagnostic studies about adults reported in the literature used the MRI as a tool to understand the presence of the ligament. The existence of the ligament is commonly described in the literature, but the identification of a tear is not fully known.

Gossner stated that the presence of the ALL can be described in the majority of patients undergoing a standard MRI of the knee [49]. The author suggested that with a decrease in cut depth, the view can be improved, and both orthopedic surgeons and radiologists must carefully review this ligament by studying MRI scans of the knee.

Similarly, Hartigan et al. reported a high recognition of this structure (100% of the 72 MRIs evaluated by two radiologists). However, ALL tears were noted in 26% by one radiologist and in 62% by the other radiologist. The authors stated using standard 1.5-tesla MRI sequences is not sufficient to detect ALL tears and concluded that accurate imaging sequences may be fundamental to understanding the presence of tears in this structure [50].

According to other authors [44], an ALL tear can be detected in only one-third of the knee MRIs of acute ACL injuries, and the tear is usually in the proximal aspect of the ligament.

Cavaignac et al. [51] reported a 100% sensitivity rate for detecting the presence of the ALL using ultrasonography; however, they did not report the accuracy of detecting the ligament tear. Kandel et al. [52] stated that ultrasonography is a reliable method for evaluating the ALL and can be used in clinical practice. No studies have been conducted so far about the detection of the ALL using ultrasonography in children.

On the other hand, all the diagnostic studies performed on pediatric patients [32,33] were conducted using MRI and only considered uninjured knees, reporting a presence of the ALL of around 70%. Liebenstein et al. [32] examined 61 MRIs from teenagers with a mean age of 15 years (±2.3) and did not find any correlation between age and the presence of the ligament. In contrast, Helito et al. found that no female patients under 7 years of age and no male patients under 6 years of age had the ALL after an evaluation of 363 MRI scans of the knee, suggesting that the ligament may develop at some point during growth [33].

The presence of the ALL has also been studied in cadaveric specimens, showing different results depending on the age of the children. Shea et al. [36] conducted a study on eight specimens and found the ligament only in one case. However, as they stated, the restricted number of specimens and especially the age of most of the specimens (mean age of 3.4 years and a range of 3 months–10 years) could be an important limitation of the study.

After a few years, the same authors [34] dissected 14 specimens with a mean age of 14 years (range of 7–11 years), and they reported the presence of the ligament in 9 out of 14 patients. A comparison of these two studies, considering the difference in the presence of the ligament in two different group ages, suggests that the ligament may become more distinct as it develops over time. The authors described that the ligament was very variable (sometimes very thin, sometimes sheet-like, and in some cases, a well-defined “pearly band”). In addition to the structure itself, the origin of the ligament also varies largely, according to the authors. Three ALL origins were located distal and anterior from their respective lateral collateral ligament origins. Occasionally, the ALL and lateral collateral ligament shared the same origin location (two cases); in some cases, the ALL origin was proximal and posterior to the lateral collateral ligament origin; in one single case, the ALL was anterior to the lateral collateral ligament; and in another case, the ALL was proximal and anterior to the lateral collateral ligament beginning.

Cadaveric studies were also performed on fetal specimens [53], as reported by Helito et al. [35], who dissected and performed a histological analysis of 20 specimens with the age of 28.64 ± 3.20 weeks. After applying an internal rotation of the tibia during knee flexion, the authors clearly identified the ligament in 20 out of 20 patients, near the LCL, just underneath the iliotibial tract. Also in this circumstance, a large variety in the origin of the ligament was reported among the different cases. The histological analysis of the ligament exhibited dense, well-organized collagenous (type I) fibers with elongated fibroblasts and an increased cell concentration compared with the adult ALL. This last study differs in a significant way from the study of Shea et al. [34,36], which suggested that the ALL is an inconstant structure in children and that it only increases after the anterolateral capsule of the knee is subjected to physiological loads. Conversely, Helito et al. [35] stated that the ALL previously appear in fetuses with a mean age of 28.64 weeks and develops similarly to other ligaments of the knee region.

More recently, Iseki et al. [31] aimed to clearly identify a distinct ligament in the anterolateral capsule (ALC) of the knee. They dissected 15 ALC specimens (aged 6.3 ± 3.3 years), 5 lateral collateral ligaments (LCLs) (aged 3.4 ± 1.3 years), and 5 quadriceps tendons (QTs) (aged 2.0 ± 1.2 years). The authors performed an RNA isolation and gene expression analysis, a histology and immunohistochemistry analysis, and a cell morphology analysis. Using those tools, a clear distinct ligament could not be observed in the ALC.

Contemplating the high variability in the anterolateral knee anatomy, in 2022, Randhawa et al. [29] examined nine pediatric cadaveric knee specimens (aged 4.2 years: range 2 months–10 years) with the aim of fully understanding the anatomy of this area. The authors mainly focused on the LCL, Popliteus, and ileo–tibial band (ITB), while the ALL was only marginal in their study.

Considering the importance of the aforementioned ligament in surgical procedures, Madhan et al. [30] wanted to understand the knowledge and preferences of surgeons in clinical practice for performing ACL reconstruction in children and adolescents together with ALL reconstruction or LET. The authors reported that in the sample of 63 surgeons engaged using a survey, around 50% of pediatric sports surgeons occasionally performed ALL augmentation with primary ACLR and 79% with revision ACLR. The authors noticed that doctors with sports medicine fellowships were more likely to execute these procedures. Almost all the surgeons who perform these techniques have explicit curiosity in investigating them prospectively or to randomize patients.

### 4.6. Limitations

The present study has some limitations. First, not many studies about the ALL in children were reported in the literature. Furthermore, the way many studies were conducted was heterogeneous, and sometimes they lead to unreliable conclusions.

## 5. Conclusions

The existence, anatomy, function, and biomechanics of the ALL are still not well-known.

The ALL is a rotational stabilizer of the knee, but its anatomy varies largely between individuals of different ages. This ligament needs to be further explored in adults and in children. Only a few articles studied the ALL in the pediatric population. Further studies on diagnostic imaging, specimens, and especially in vivo patients are necessary to obtain detailed knowledge of the ALL.

## Figures and Tables

**Figure 1 jfmk-08-00126-f001:**
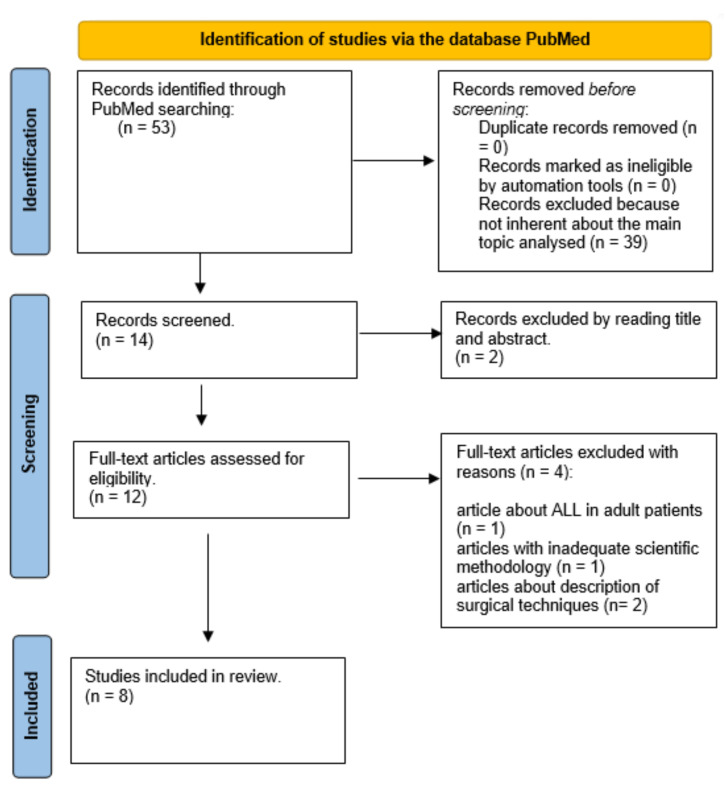
Identification of studies.

**Table 1 jfmk-08-00126-t001:** Main results of the eight selected articles.

Author	Year	Article	Aim of the Study	Sample	Control Group	Female	Male	Right	Left	Mean Age	Results	Type of Study
Randhawa [29]	2022	Pediatric Reference Anatomy for ACL Reconstruction and Secondary Anterolateral Ligament or Lateral Extra-Articular Tenodesis Procedures.	To evaluate the structure of the knee joint physes, lateral collateral ligament (LCL) origin, popliteus origin, and ITB attachment.	9 cadaveric	-	7	2	5	4	4.2 (range 10 months–10 years)	It explains positions of the femoral lateral collateral ligament (LCL), popliteus origins and tibial iliotibial band (ITB) attachment, and their own physeal relations.	Cadaveric/descriptive anatomic study
Madhan [30]	2022	Trends in Anterolateral Ligament Reconstruction and Lateral Extra-articular Tenodesis with ACL Reconstruction in Children and Adolescents.	To measure physician methods for anterolateral ligament reconstruction (ALLR) and lateral extra-articular tenodesis (LET) in pediatric patients.	63 surgical	-	-	-	-	-		More than 50% of pediatric sports surgeons occasionally perform ALL augmentation with primary ACLR and 79% with revision ACLR. Doctors with sports medicine fellowships were more likely to execute these procedures.	Survey— cross-sectional study.
Iseki [31]	2021	Paediatric Knee Anterolateral Capsule Does Not Contain a Distinct Ligament: Analysis of Histology, Immunohistochemistry and Gene Expression.	To explore the existence of the ligament phenotype in the pediatric knee anterolateral capsule (ALC).	15 fresh cadaveric specimens	5 pediatric LCLs (age 3.4 ± 1.3 years) 5 pediatric quadriceps tendon QTs (age 2.0 ± 1.2 years)					6.3 ± 3.3 years	A clear ligament could not be detected in the ALC established using histology, immunohistochemistry, and gene expression evaluation.	Controlled laboratory study
Liebensteiner [32]	2019	The Anterolateral Ligament and the Deep Structures of the Iliotibial Tract: MRI Visibility in the Paediatric Patient.	To evaluate the view of the ALL and the deep structures in theiliotibial tract using the MRI in pediatric patients.	61 patients	-	36	25			15 years (±2.3)	ALL (and the deep structures of the ITT) can be detected using MRI.	Diagnostic study—level 3.
Helito [33]	2018	Magnetic Resonance Imaging Assessment of the Normal Knee Anterolateral Ligament in Children and Adolescents.	To describe the ALL in healthy knees of pediatric patients using MRI.	363 patients		163	200				Vision of the ALL improves with age.	Diagnostic study—Level 3
Shea [34]	2017	Anterolateral Ligament of the Knee Shows Variable Anatomy in Pediatric Specimens.	To explore the existence of the ALL in preadolescent anatomic specimens.	14 cadaveric specimens		2	12	7	7	8 years (range 7–11 years)	The occurrence of the ALL in pediatric specimens is lower thanin adult specimens.	Cadaveric study
Helito [35]	2017	Anterolateral Ligament of the Fetal Knee: An Anatomic and Histological Study.	To assess the ALL in human fetuses and establish its existence.	20 fetal cadaveric specimens		10	10	10	10	28.64 ± 3.20 weeks.	The ALL is present during fetal development.	Descriptive laboratory study
Shea [36]	2016	The Anterolateral Ligament of the Knee: An Inconsistent Finding in Pediatric Cadaveric Specimens.	To estimate whether the ALL might be detected in pediatric cadaveric knee specimens.	8 cadaveric specimens		5	3	7	1	3.4 years (range 3 months–10 years)	The ALL could grow late in life.	Cadaveric study

## Data Availability

Not applicable.

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
