# Peer review of "The Anterolateral Ligament of the Knee in Pediatric Patients: What Do We Know? A Scoping Review"

_jfmk, 2023, doi:10.3390/jfmk8030126_

Round 1

Reviewer 1 Report

The paper presented for the review is aimed to analyse all the literature available about ALL in pediatric patients.

The authors raised the interesting topic of an unusual anatomical phenomenon which is uncertain in children in terms of both identifying and understanding of the clinical meaning. It is emphasised that the existence of the ligament and it's prevalence is unclear in pediatric patients. To clarify these questions an attempt of the systematic review was done. The systematic search according to general principles of PRISMA was done. As the result of the search 8 papers were found and analysed. Despite the relatively small number of the sources it would be still enough for the following analysis if the data would be more consistent and more informative. In fact, among those 8 papers 5 items are pure anatomical (cadaveric and fetal autopsy), 2 are relatively anatomical (related to the MRI anatomy), and 1 is the survey of the orthopaedic surgeons about the general impression regarding ALL. Among these anatomical studies, both direct and MRI, which (as it was mentioned cover all, but one analysed papers) the main research question is roughly the existence and anatomical detectability of the ligament in children. 

If we just look at the amount of the text regarding the different domains of the study it is easy to see that the absolute majority of the study is related to the general, but not pediatric issues. The discussion part  (which is the most important for the systematic review) the majority of the text is related to the adults. Even in the chapter “4.5 Overview on pediatric population” only few paragraphs are relevant to the pediatrics. And the most of the cited and discussed papers both in the introduction and discussion are related to the adults (especially regarding surgical procedures).

The main point of the systematic review is not just applying the systematic approach to the literature search but first of all using the methodology of systematic analysis of the results of this search. The authors provided the first part of this principle but not the second. As the result - if we take the title of the paper “The anterolateral ligament of the knee in paediatric patients: what do we know?” - the answer from the paper is “almost nothing”…

Meanwhile the major advantage of the study is clear - the authors raised the question of the anatomy of the ligament or more specifically - if the ligament does exist as a separate anatomical structure in children and why the presence of the ligament is much less common in pediatric patients? Actually all but one informatively doubtful paper among those found with the systematic search are related to this issue. 

In conclusion it is recommended to redesign the paper according to the content. Since the analysed papers are not consistent in terms of the material and the research question of the study it is possible to determine the paper as the narrative, but not systematic review. Otherwise it is possible to focus on the anatomical aspects and rearrange the analytical parts with systematic analysis of the material.

Author Response

Thank you very much for your comments, we really appreciate that. If you agree, we can consider the article as a scoping review. We did the necessary changes.

Reviewer 2 Report

The study contributes novel facts to the research field analysing the presence of the ALL in children. I like the information about development (p12). Are there known facts concerning the relation of the ALL to epiphyseal plates around the knee? Several statements lack supporting references. Examples are listed below. The table needs a better formate, since it is not really readable in the presented manner. it would be nice to add to the cadaveric specimen whether they are fixed (how?) or not (fresh frozen). A couple of sentences should be rewritten to improve understanding and some inconsistencies in writing style should be adapted. If possible, a sketch of the ALL would be helpful.

fig. 1: left site: write vertical

line 53: insert blank before citations

table column "sample" second raw "surgeons" should be substituted by "surgical", since it is headed "sample" the word "specimen" could be omitted

line 133: the abbreviation ALC should be inserted after capsule

line 157: "ITT"  perhaps not explained? does it mean iliotibial tract?

line 168: "Vincent" add "et al."

next line: write "anatomical" compare line 223

line 185-187: needs citation

line 202: better "that it is possible"

line 208: restoring

line 211: also in adolsecents?

line 214, 223 and 229-230, 239, 279: add references

line 220: add point

line 230: add references or refer to the table

line 241, 257, 279, 281: abbreviate anterolateral ligament

line 243: write "can be improved"

line 244: could [47] be added at the end of the sentence? and [48] in 250?

line 261: substitute "conducted" by "performed" otherwise it is 2x used in the same sentence

line 262: et al., (point)

lines 264-266: rewrite sentence

line 281/301 etc lateral collateral ligament, use the abbreviation consistently, compare with line 174 FCL (175, 179), it seems to be the same

line 308 "ITB" introduce it earlier

lin313: LET has alrady been introduced

4.6 what is the influence of fixation on the presence of the ALL in cadaveric samples, should it be mentioned in the table whether the samples were fixed or not?

some sentences should be rewritten for better understanding (see above)

Author Response

The study contributes novel facts to the research field analysing the presence of the ALL in children. I like the information about development (p12). Are there known facts concerning the relation of the ALL to epiphyseal plates around the knee? Several statements lack supporting references. Examples are listed below. The table needs a better formate, since it is not really readable in the presented manner. it would be nice to add to the cadaveric specimen whether they are fixed (how?) or not (fresh frozen). A couple of sentences should be rewritten to improve understanding and some inconsistencies in writing style should be adapted. If possible, a sketch of the ALL would be helpful.

Thank you very much for the comment. We appreciate. Unfortunately, there are not information about the relation of the ALL to epiphyseal plates. Furthermore, many articles do not report how the cadaveric specimens were fixed: we wrote this information in the table when the authors reported it. As you said, the table is a little bit irregular, we totally agree with that. At this point is not possible to change it without compromising it, but we are confident this graphic issue can be easily fixed during the final layout.

fig. 1: left site: write vertical

Response 2: Thank you very much. We modified it

line 53: insert blank before citations

Thank you very much. We did it

table column "sample" second raw "surgeons" should be substituted by "surgical", since it is headed "sample" the word "specimen" could be omitted

Thank you very much. We changed it

line 133: the abbreviation ALC should be inserted after capsule

Thank you very much. We changed it

line 157: "ITT" perhaps not explained? does it mean iliotibial tract?

Thank you very much. We added the explanation.

line 168: "Vincent" add "et al."

Thank you very much. We added it

next line: write "anatomical" compare line 223

Ok, we wrote it

line 185-187: needs citation

Thanks. We added it

line 202: better "that it is possible"

Thanks. We changed it

line 208: restoring

Thanks. We changed it

line 211: also in adolsecents?

Yes

line 214, 223 and 229-230, 239, 279: add references

Done

line 220: add point

Done

line 230: add references or refer to the table

Done

line 241, 257, 279, 281: abbreviate anterolateral ligament

Done

line 243: write "can be improved"

Done

line 244: could [47] be added at the end of the sentence? and [48] in 250?

Done

line 261: substitute "conducted" by "performed" otherwise it is 2x used in the same sentence

Done

line 262: et al., (point)

Done

lines 264-266: rewrite sentence

Done

line 281/301 etc lateral collateral ligament, use the abbreviation consistently, compare with line 174 FCL (175, 179), it seems to be the same

Thanks, we changed FCL to LCL

line 308 "ITB" introduce it earlier

Done

lin313: LET has alrady been introduced

We removed it, thanks

4.6 what is the influence of fixation on the presence of the ALL in cadaveric samples, should it be mentioned in the table whether the samples were fixed or not?

As said before, unfortunately, most of the articles did not report this information

Round 2

Reviewer 1 Report

The authors followed the reviewer’s comments and modified the paper according the recommendations. The paper is recommended for publication in the present version.

Reviewer 2 Report

My comments have been fully adressed.